# Evaluation of Organic and Synthetic Herbicide Applications on Weed Suppression in a Conventional Cropping System in Louisiana

Caitlin deNux [1,*], Aixin Hou [2] and Lisa Fultz [3]

1    School of Geoscience, University of Louisiana at Lafayette, 611 McKinley St., Lafayette, LA 70504, USA
2    College of the Coast & Environment, Louisiana State University, 93 South Quad Drive, Baton Rouge, LA 70803, USA; ahou@lsu.edu
3    School of Plant, Environmental and Soil Sciences, Louisiana State University AgCenter, 104 M.B. Sturgis Hall, Baton Rouge, LA 70803, USA; lfultz@agcenter.lsu.edu
*    Correspondence: caitlin.woodard@louisiana.edu

**Abstract:** Synthetic herbicides, with their varying modes of action, are well known for their efficiency in the suppression and control of weed species in U.S. agriculture. However, the consequences of using synthetic herbicides without attention to the surrounding environment produce chemical run-off, changes in soil health and soil health conditions, and create herbicide-resistant weeds. These outcomes have encouraged growers to seek alternative methods for their weed management programs or farming operations. Organic production systems and organic pesticides have helped address these challenges related to sustainability and environmental health. However, the use of organic herbicides in a conventional cropping system is not usually evaluated, as the effectiveness of these organic herbicides on weed populations in such a setting is thought to be inferior when compared to their synthetic counterparts. In this study, organic and synthetic herbicides were assessed on their performance in weed suppression surveys. The experimental design included nine treatments with four replications on two different soil types. The results showed organic herbicides were not comparable to synthetic herbicides in weed suppression. In weed management programs, using recommended herbicide application rates outlined on the herbicide label and conducting applications with environmental stewardship in mind could decrease possible herbicide effects within the environment.

**Keywords:** organic; herbicide; sustainable agriculture; weed suppression

## 1. Introduction

A weed is defined as any plant that is considered undesirable in a particular location and can typically be classified into either grasses, sedges, or broadleaf weed species [1,2]. When evaluating morphological characters, weeds that are classified as grasses are considered monocots and belong to the family *Poaceae*. Characteristically, they have long, narrow, spiny leaves with parallel veins and can have an annual or perennial life cycle. Those classified as sedges are also considered monocots and belong to the family *Cyperaceae*. Their leaves form at the base of the plant, have modified triangular-shaped stems, and can have an annual or perennial life cycle. Those classified as broadleaf weed species encompass many of the remaining plant families and are dicot plants. These weed species have wider leaves than grasses, branched stems, and netlike net venation characteristics and can have an annual, biennial, or perennial life cycle [3]. Annual weed species live for one growing season and are often referred to as either summer or winter annuals. Biennial weed species live up to two years, with seeds germinating in the spring, summer, or fall of the first year and repopulating during the summer of the second year, and perennial weed species are those that produce vegetative structures that allow them to live for three or more years [3].

The presence and competition of weed populations in agricultural fields can negatively impact the cash crop by lowering the value of the harvested crop, reducing protein in grains, and decreasing fruit or seed size in desired crops. Before the introduction of herbicides in the 1950s, most weed control methods in cropping systems were accomplished with a combination of hand weeding, the use of mechanical cultivation tools, and cultural practices [4]. Weed control during that period was time-consuming and a major task for producers. As synthetic herbicides became more common after World War II, producers found this chemical control method to be highly effective against target organisms, reduced labor costs, reduced soil erosion, and exhibited a higher level of control than cultivation tools. Modern agriculture now uses a variety of synthetic herbicides with varying modes of action to control weed populations. Today, competition from weed species accounts for 34% of the crop loss that occurs in agricultural fields worldwide [5]. As a result, herbicide expenditures account for two-thirds of all pesticides used in US agriculture as growers try to control weeds in their operations [4].

The efficacy of herbicides can be influenced by many factors, the most important being environmental [6]. Herbicide solubility, movement, and degradation within the soil are directly affected by soil temperature and moisture. Soil water and temperature can also affect herbicide behavior through changes in plant root characteristics, permeability, and transport through transpiration flow. Dry conditions can reduce the efficiency of herbicides. Some herbicides require soil moisture to be present prior to spray applications for greater activity (i.e., Prowl $H_2O$). Air temperature and humidity affect evaporation and volitation, decreasing the amount of herbicide on target plants and active sites, reducing droplet size, and increasing opportunities for herbicide drift. Morphological changes in plant species caused by climate change or increased levels of elevated $CO_2$ levels may also have an effect on herbicide efficacy [7]. These can be seen as variations in enzymatic activity, pigment production, increased starch levels in C3 plants, and decreased protein levels. All of which have been shown to interfere with herbicide activity. These influences are important as they help dictate herbicide persistence and how effective an herbicide may be over weed species populations.

The success of synthetic herbicides grew in part due to the introduction of glyphosate-resistant (GR) crops in 1966 [8]. With this technology, growers can make a single application of glyphosate at almost any time to control all weed species and weed populations. This eliminated the need for additional herbicides and herbicide applications, was more cost-efficient, and provided opportunities for higher-yielding crops. When available, growers rapidly adopted GR crops into their cropping systems, and weed management programs became focused on glyphosate herbicide applications for control. However, the extended use of glyphosate alone in GR crops led to the overuse of this herbicide in most fields and contributed to the widespread evolution of GR weed resistance [8]. To date, 41 weed species have been confirmed as glyphosate-resistant worldwide, including 18 weed species in North America, with half evolving in GR crop fields [6]. These include weeds such as palmer amaranth (*Amaranthus palmeri*), common ragweed (*Ambrosia artemisiifolia*), giant foxtail (*Setaria faberi*), Johnsongrass (*Sorghum halepense*), and barnyardgrass (*Echinochloa crus-galli*). Growers are now encouraged to diversify and mix herbicides to mitigate the spread of GR weeds [9].

Conventional agriculture has faced a variety of serious challenges in the past few decades. Climate change, a high rate of biodiversity loss, herbicide-resistant weeds, land degradation, compaction, pollution, rising production costs, and a decreasing number of small farms have led many to implement more sustainable, organic management systems in their operations in order to contend with growing public discourse and these environmental pressures [10]. Although there is no unified definition for sustainable agriculture due to its complex nature, many would say it could be defined as an integrated system of plant and animal practices that satisfy human food and fiber needs, enhance environmental quality, make efficient use of non-renewable resources and on-farm resources, and integrate appropriate natural biological cycles and controls [11]. The publication of Rachel Carson's

book *Silent Spring* in 1962 is seen as the beginning of the modern organic era in the United States [4]. It helped spur a consciousness for food production processes and environmental issues that have evolved and sustained through today. An increase in customer awareness of pesticide residues, food safety, human health, food quality, crop production practices, and environmental stewardship increased the demand for available organic foods. The popularity of organic farming grew with this demand. Organic agriculture relies on ecological processes, biodiversity, and management adapted to local conditions rather than using inputs with potential effects.

The introduction of organic herbicides into weed management programs is still a relatively new concept in agriculture production systems today. The United States Department of Agriculture-National Organic Program (USDA-NOP) standards relating to weed control require organic producers to use cultural and physical tools before using any approved pesticides [4]. Organic cropping systems revolve around using a range of techniques within a natural-based system instead of a single control method for managing weed populations. The inclusion of cover crops, crop rotation, mulch, and manures are used interchangeably and in combination to provide an effective weed control system. Organic herbicides are non-selective, and those available use a variety of natural active ingredients in their chemical formulations. These can range from natural chemicals to essential oils. Some of the most common organic active ingredients used today are corn gluten meal (CGM), mustard meal (MM), vinegar (5%, 10%, and 20% acetic acid), clove oil, ammonium nonanoate, 55% d-limonene, lemongrass oil, and pelargonic acid [12]. Corn gluten meal, a by-product of the wet-milling process of corn, and MM are phytotoxic. These two active ingredients inhibit root development, decrease shoot length, and inhibit the plant survival of weed and crop seedlings. CGM and MM are effective against established turf, transplanted vegetables, and direct-seeded vegetables. Products containing vinegar (i.e., acetic acid) are more effective in controlling grasses and annual weed species than biennial and perennial broadleaf weeds. Weed control is more effective with higher rates of acetic acid and application volume.

Clove oil applied at lower application volumes provides weed control comparable to acetic acid, with broadleaf control greater than grass control. Adding adjuvants can increase weed control efficiency for herbicides with clove oil. Ammonium nonanoate forms from the biodegradation of higher fatty acids and is the most effective on broadleaf weeds and small, young weeds. Lower applications of ammonium nonanoate can be more effective for weed control than acetic acid products. D-limonene at 55% acts as a degreasing agent, essentially dissolving the waxy cuticle on plant tissues. Plants with thinner cuticles are the most responsive to these herbicides, and young, small weeds are the most susceptible. Lemongrass oil disrupts the polymerization of plant microtubules. Herbicides with this active ingredient are contact herbicides, and portions of plants receiving the spray solution will be affected. Pelargonic acid is used in contact with burndown herbicides, inhibiting cell membranes by causing cell leakage and membrane lipid breakdown. The use of these herbicides has the potential to provide a safe, food-grade, non-toxic herbicide alternative that would have limited lasting effects on its surrounding environment.

Limited information is available in the literature regarding organic herbicides' effect on suppressing weed populations. Articles refer to a complex agroecosystem approach when referencing weed management programs and organic agriculture. Mechanical tillage, direct physical weed control methods, crop fertilization strategies, crop rotation, intercropping, irrigation, and cover crops are used together in an organic weed management system, but herbicides are rarely mentioned [13]. As a result, commercially available organic herbicides in the market today are primarily rated for residential and small operation weed management programs. This relays the importance of this study, as organic herbicide efficiency has not been tested in a conventional cropping system, nor have these specific herbicides been evaluated on weed suppression.

The objective of this study was to evaluate weed suppression against natural and synthetic herbicide applications. As organic herbicides have the potential to affect plants

through natural pathways, it was hypothesized that these treatments would be comparable to synthetic treatments in weed suppression and control.

## 2. Materials and Methods

### 2.1. Site Characteristics

A two-year non-irrigated experiment was conducted at the Louisiana State University Agricultural Center's H. Rouse Caffey Rice Research Station's South Farm, located 5.96 km south of Crowley, Louisiana, from 2020 to 2022. An additional location was added in 2022 on a producer's cultivated field located 6.76 km northeast of Crowley, Louisiana (Figure 1).

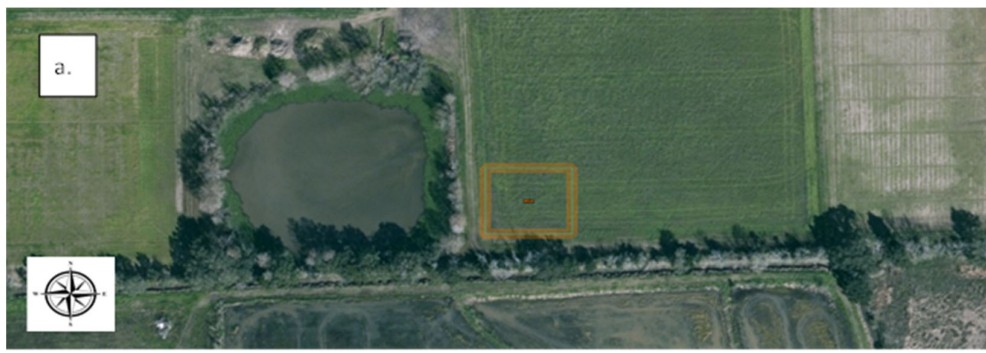

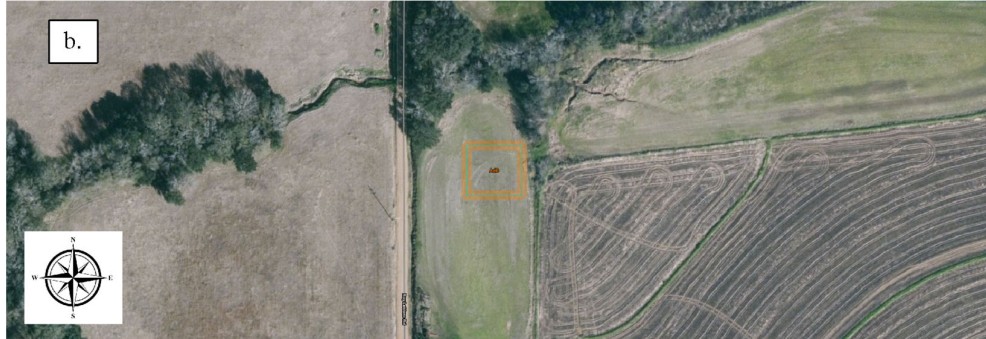

**Figure 1.** Aerial view of (**a**) Midland silty clay loam (latitude, longitude: 30.175246° N, −92.358995° W) and (**b**) Acadiana silt loam (latitude, longitude: 30.251687° N, −92.341026° W) locations for 2021 and 2022.

Field locations included two areas with soil textures classified as silt loam and very fine sandy loam, specifically Midland silty clay loam (MdA) and Acadiana silt loam (AdB) soils. These were located approximately 14.5 km apart. The MdA soil was classified as a fine, smectitic, thermic Chromic Vertic Epiaqualfs and a very deep, poorly drained, slowly permeable soil [14]. The AdB is classified as a fine, mixed, active, thermic Vertic Paleudalfs and is a very deep, moderately well-drained, slowly permeable soil [15]. The previous crop planted was soybean (*Glycine max*).

### 2.2. Experimental Design and Field Management

The experimental design for this study was a randomized complete block (RCB) design with each plot approximately 0.00056 ha$^{-1}$ in size. Treatments were randomly assigned to each replication. Initial field preparation was prepared with a John Deere 7750 tractor (Deere & Company, Waterloo, IA, USA) with a 4.6 m Landoll plow at about 12.7 cm in depth. This was conducted to incorporate the previous year's vegetative residue for each site during the fall season prior to treatment applications. This study included nine treatments consisting of four organic herbicides and four conventional synthetic herbicides, in addition to a control plot where no treatments were applied (Table 1). Application rates were determined based on label recommendations for each treatment at 57 L per hectare

for Year 1 at 1.0 L ha water per treatment. Application rates were modified in Year 2 based on efficacy measurements made in Year 1 and additional research [12,13,16]. According to the literature, repeated spray applications, specific temperature requirements, high spray volume requirements, and controlling weeds during their early growth cycle should be considered when trying to increase the efficacy of organic herbicides on weed populations. In Year 2, synthetic treatments were applied at 57 L per hectare, and organic treatments were increased at 114 L per hectare at 1.0 L ha water per treatment compared to the 57 L per hectare of the previous year. Treatments were applied with a $CO_2$ backpack sprayer and a three-tip (TJ8002) hand wand. For Year 1, treatment applications were applied on 27 May 2021. For Year 2, the number of applications was increased to improve the efficiency of the organic herbicide applications, which were applied based on weed control evaluations for each treatment (Table 2). With this in mind, herbicide treatment applications were adjusted in order to correctly evaluate organic herbicides in controlling target organisms. Regarding the dosage and organic treatments, Figure 2 presents the steps taken in Year 1 and Year 2 in regard to biomass collection, additional herbicide applications, and timed evaluations.

**Table 1.** Herbicide treatments based on label recommendation rates at kg/ha$^{-1}$. Treatment number, type of herbicide, chemical family, and active ingredients of organic and synthetic herbicides are presented.

| # | Type | Chemical Family | Active Ingredient | Treatment | Application Rate (kg ha$^{-1}$) |
|---|---|---|---|---|---|
| 1 | - | - | - | Control | - |
| 2 | Organic | - | Sodium Chloride | DIY: Salt | 6.80 |
|   |   | - | Triclosan | Dish Soap | 0.85 |
|   |   | - | 5% Acetic Acid | Vinegar | 54.43 |
| 3 | Organic | N/A | 20% Acetic Acid | Vinegar Weed & Grass Killer | 54.23 |
| 4 | Organic | N/A | d-limonene 55% | Avenger Weed Killer Concentrate | 10.89 |
| 5 | Synthetic | Dinitroaniline | Pendimethalin | Prowl $H_2O$ | 1.36 |
| 6 | Synthetic | Glycine | Glyphosate | Roundup | 1.06 |
| 7 | Synthetic | Benzoic Acid | Dicamba | Rifle | 0.45 |
| 8 | Organic | Fatty Acid | Pelargonic Acid | Scythe | 2.75 |
| 9 | Synthetic | Iron Salts | Iron EDTA | Fiesta | 2.13 |

**Table 2.** Year 2 treatment application dates by site location. The table includes applied treatment applications for each additional application needed.

| Date | Sites | Treatments |
|---|---|---|
| 28 April 2022 | 1 and 2 | 1–9 |
| 11 May 2022 | 1 | 1, 2, 3, 4, 5, 7, 8, 9 |
| 14 June 2022 | 1 | 1–9 |
| 5 May 2022 | 2 | 1, 2, 3, 4, 5, 8, 9 |
| 3 June 2022 | 2 | 1–9 |

Initial treatments (organic and synthetic herbicides) were applied on 28 April 2022, when vegetation populations were 15.2 cm or below in height and field conditions were dry. Additional applications in Year 2 were applied based on weed control evaluations, temperature, and soil moisture (Figure 3). Average temperatures and annual rainfall for both site locations were similar but not identical. Climate data were recorded near site 2 and Louisiana State University Agricultural Center's H. Rouse Caffey Rice Research Station's North Farm. Site 1 did not have a nearby weather station provided for these climate data. However, as both locations were relatively close by, it can be inferred that both locations had similar climates.

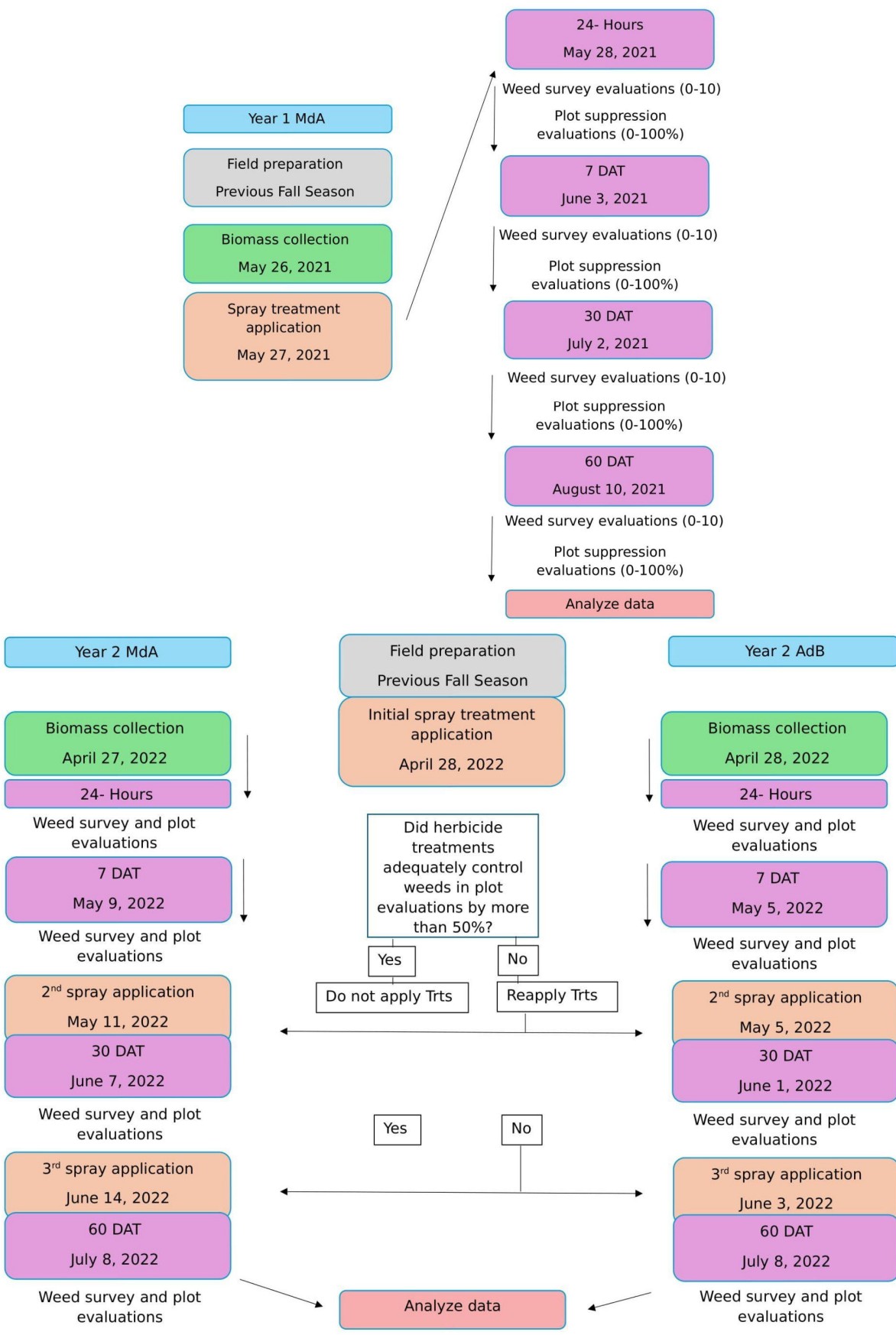

**Figure 2.** Flowchart showcasing timed evaluations, biomass collection, and spray application during 2021 and 2022 for site 1 and site 2.

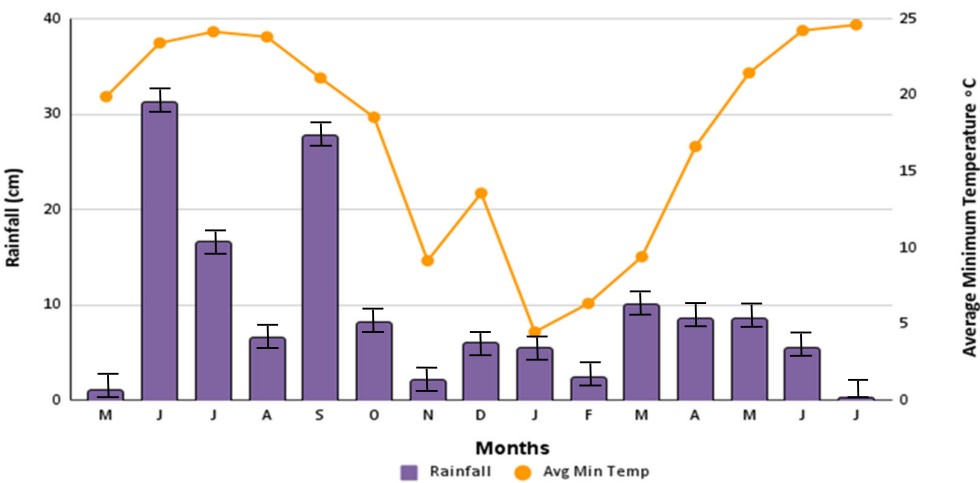

**Figure 3.** Monthly precipitation and average minimum temperatures recorded for MdA and AdB sites from 2021 to 2022 at the H. Rouse Caffey Rice Research Station Weather Station, Rayne, Louisiana.

*2.3. Weed Surveys and Weed Biomass*

A weed survey is a process of visually evaluating weed species in a survey setting that can be repeated or data that are collected over a longer period and can be used to quantify changes in species abundance and diversity [17]. Weed surveys were conducted, and weed biomass samples were taken to evaluate how effective the treatments were in controlling targeted populations over two months. Moisture content was calculated based on weight loss results. Samples were collected using a 50 cm square made of PVC pipe prior to the first herbicide application for plant cover evaluations. Weed species for Year 1 (Table 3) and Year 2 (Table 4) can be found below.

**Table 3.** Year 1 weed species for site 1 (MdA). The table includes weed species, location of weed species, scientific name, leaf type, and growing season of weed populations.

| Common Name | Scientific Name | Growth Forms | Life Cycle |
| --- | --- | --- | --- |
| Yellow nutsedge | *Cyperus esculentus* | Sedge | Perennial |
| Lawn burweed | *Soliva sessilis* | Broadleaf | Annual |
| Blue water hyssop | *Bacopa caroliniana* | Broadleaf | Perennial |
| Doveweed | *Murdannia nudiflora* | Broadleaf | Annual |
| Barnyardgrass | *Echinochloa* | Grass | Annual |
| Common yellow woodsorrel | *Oxalis stricta* | Broadleaf | Perennial and Annual |
| Purple cudweed | *Gamochaeta purpurea* | Broadleaf | Annual and/or Biennial |
| Spreading hedge | *Torilis arvensis* | Broadleaf | Annual |
| Narrowleaf aster | *Sericocarpus linifolius* | Grass | Perennial |
| Japanese mazus | *Mazus pumilus* | Broadleaf | Annual |
| Dog fennel | *Eupatorium capillifolium* | Broadleaf | Perennial |
| Alligator weed | *Alternanthera philoxeroides* | Broadleaf | Perennial |
| Dallisgrass | *Paspalum dilatatum* | Grass | Perennial |
| Grasslike fimbry | *Fimbristylis miliacea* | Sedge | Annual |
| Chickweed | *Stellaria media* | Broadleaf | Annual |
| Hedge hyssop | *Gratiola viscidula* | Broadleaf | Perennial |
| Valley redstem | *Ammannia coccinea* | Broadleaf | Annual |

**Table 4.** Year 2 weed species information for site 1 (MdA) and site 2 (AdB). The table includes weed species, location of weed species, scientific name, leaf type, and growing season of weed populations in 2022.

| Common Name | Site | Scientific Name | Growth Forms | Life Cycle |
|---|---|---|---|---|
| Yellow nutsedge | MdA | *Cyperus esculentus* | Sedge | Perennial |
| Lawn burweed | MdA/AdB | *Soliva sessilis* | Broadleaf | Annual |
| Blue water hyssop | MdA/AdB | *Bacopa caroliniana* | Broadleaf | Perennial |
| Doveweed | MdA/AdB | *Murdannia nudiflora* | Broadleaf | Annual |
| Jungle rice | AdB | *Echinochloa colona* | Grass | Annual |
| Common yellow woodsorrel | MdA/AdB | *Oxalis stricta* | Broadleaf | Annual or Perennial |
| Purple cudweed | MdA/AdB | *Gamochaeta purpurea* | Broadleaf | Annual and/or Biennial |
| Spreading hedge | MdA | *Torilis arvensis* | Broadleaf | Annual |
| Iva annua | MdA/AdB | *Iva Annua* | Broadleaf | Annual |
| Tall goldenrod | MdA/AdB | *Solidago canadensis* | Broadleaf | Perennial |
| Horseweed | AdB | *Erigeron canadensis* | Broadleaf | Annual and/or Biennial |
| American burnweed | AdB | *Erechtites hieraciifolius* | Broadleaf | Annual |
| Dog fennel | MdA | *Eupatorium capillifolium* | Broadleaf | Perennial |
| Alligator weed | MdA | *Alternanthera philoxeroides* | Broadleaf | Perennial |
| Dallisgrass | MdA | *Paspalum dilatatum* | Grass | Perennial |
| Grasslike fimbry | MdA | *Fimbristylis miliacea* | Sedge | Annual |
| Southern crabgrass | AdB | *Digitaria ciliaris* | Grass | Annual |
| Dandelion | MdA/AdB | *Taraxacum* | Broadleaf | Perennial |
| Spiny sowthistle | MdA/AdB | *Sonchus asper* | Broadleaf | Annual |
| Spotted spurge | AdB | *Euphorbia maculata* | Broadleaf | Annual |
| Chickweed | MdA/AdB | *Stellaria media* | Broadleaf | Annual |
| Hedge hyssop | MdA | *Gratiola viscidula* | Broadleaf | Perennial |
| Narrowleaf aster | MdA | *Sericocarpus linifolius* | Grass | Perennial |
| Fox tail | MdA | *Alopecurus* | Grass | Annual or Perennial |
| Western buttercup | MdA/AdB | *Ranunculus occidentalis* | Broadleaf | Perennial |
| Burr clover | AdB | *Medicago polymorpha* | Broadleaf | Annual |
| Pokeweed | MdA | *Phytolacca americana* | Broadleaf | Perennial |
| White clover | AdB | *Trifolium repens* | Broadleaf | Perennial |
| Grassleaf rush | AdB | *Juncus marginatus* | Rush | Perennial |
| Pink evening primrose | AdB | *Oenothera speciosa* | Broadleaf | Perennial |
| Longleaf wedgescale | AdB | *Sphenopholis filiformis* | Grass | Perennial |
| Paraguayan windmill grass | AdB | *Chloris canterai* | Grass | Perennial |

Green biomass samples were weighed on a small scale and allowed to dry in open air for two months to remove all moisture from the sample. Dry samples were then weighed again, and weights were recorded after two months. Weed surveys were conducted five times throughout the study. This included twenty-four hours after the herbicide application, 7 days after application, 30 days after application, and 60 days after application (Table 5).

**Table 5.** Year 1 and Year 2 weed biomass and weed survey evaluation dates 24 hours, 7 days, 30 days, and 60 days after treatment (DAT) by site location.

| Site | Biomass Collection | 24 H | 7 DAT | 30 DAT | 60 DAT |
|---|---|---|---|---|---|
| MdA | 26 May 2021 | 28 May 2021 | 3 June 2021 | 2 July 2021 | 10 August 2021 |
| Year 2 | | | | | |
| MdA | 27 April 2022 | 29 April 2022 | 9 May 2022 | 7 June 2022 | 8 July 2022 |
| AdB | 28 April 2022 | 29 April 2022 | 5 May 2022 | 1 June 2022 | 8 July 2022 |

### 2.4. Linear Rating Scale for Weed Control

Herbicide efficacy for weed control and suppression was evaluated for each weed species. This was achieved using a 0–10 rating scale, with zero indicating no damage and ten indicating complete weed destruction (Table 6). This is a cost-efficient, common method for rating herbicide toxicity [7]. By the terminology used in the linear rating scale, a symptom can be defined as typical discoloration, leaves turning to abnormal shapes and sizes, or stems becoming flattened; damage is any adverse, undesired effect on a plant that has been exposed to an herbicide [18]; and suppression refers to the reduction of weed biomass in each of the experimental plots [19]. Plot suppression percentages were evaluated for each experimental plot for overall weed control and suppression. This was achieved using a 0–100 rating scale, with zero indicating no control and one hundred indicating complete weed destruction for each plot. The purpose of using a weed species rating and a plot suppression percentage was to evaluate the herbicide effect on each weed species and the overall efficiency of each treatment plot.

**Table 6.** Linear rating scale used to assess herbicide phytotoxicity (0–10) across years, with 0 indicating a healthy plant and 10 representing complete weed destruction in weed species.

| Rating | Level of Weed Control |
| --- | --- |
| 0 | No damage/healthy plant |
| 1 | Poor, minute symptoms |
| 2 | Very slight symptoms, weak suppression |
| 3 | Slight but clearly visible symptoms |
| 4 | Many visible symptoms |
| 5 | Severe symptoms |
| 6 | Severe damage, moderate suppression |
| 7 | Severe damage, less than satisfactory suppression |
| 8 | Severe damage, satisfactory suppression |
| 9 | Severe damage up to complete destruction |
| 10 | Complete weed destruction |

### 2.5. Statistical Analysis

One of the most common statistical methods used in crop research is the analysis of variance (ANOVA). One-way ANOVAs are utilized when comparing the means of more than two groups when there is one independent variable [20]. Two-way ANOVA is used to test how two different independent variables, in combination, could affect a dependent variable [21]. To explore the effects of herbicide applications on weed suppression, with a focus on herbicide efficacy overall and on a per-species basis, a two-way analysis of variance (ANOVA) was utilized. In this study, dependent variables included all weed species, damage percentage, and standing biomass. Independent variables included year, treatment, site, and timing. Replication was considered a random effect. Treatment efficiency on weed suppression was analyzed using SPSS (IBM SPSS Statistics 27 Software, Armonk, NY, USA), and means were separated using Fisher's least significant difference with the LSD option of the MEANs statement. An $\alpha \leq 0.05$ was considered significantly different for all procedures.

## 3. Results

### 3.1. Weed Biomass

Weed biomass was not affected by treatments in Year 1 ($p = 0.904$) or Year 2 ($p = 0.911$). The greatest amount of standing weed biomass was measured at site 2 (41 kg ha$^{-1}$) in 2022, while the lowest biomass was measured at site 1 in 2022 (24 kg ha$^{-1}$) (Figure 4). Differences in biomass per site could have had a significant effect on herbicide efficiency. Increased biomass, particularly belowground, may have a dilution effect on herbicide efficacy, and higher weed densities can make post-emergent herbicides more likely to fail [7]. This may

occur due to overlapping leaves preventing sufficient spray coverage across target species or through less overall leaf coverage to intercept the applied herbicide.

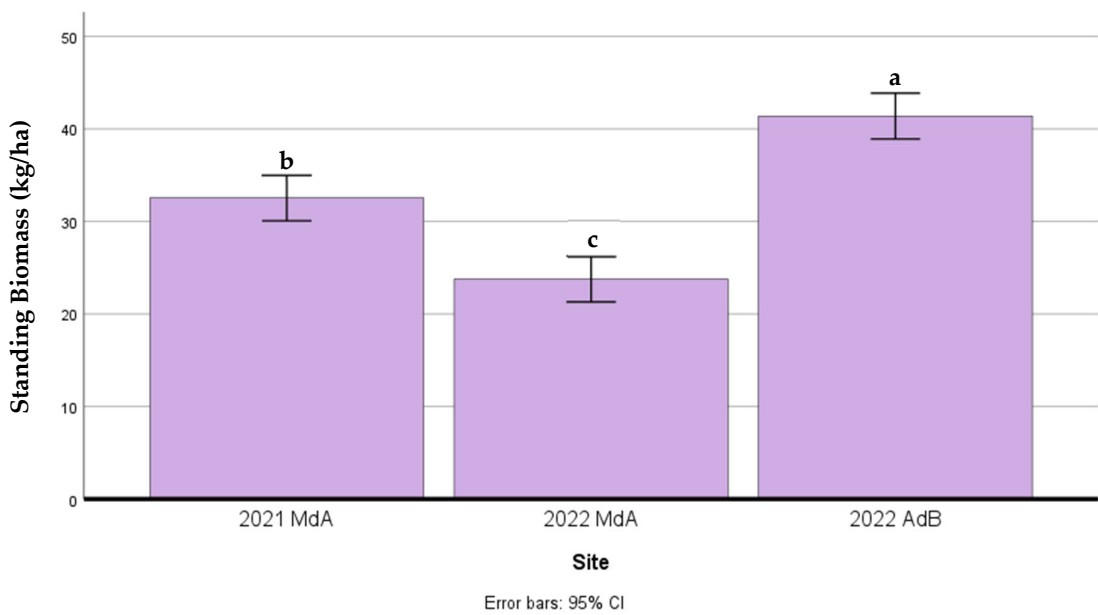

**Figure 4.** Average standing biomass for site 1 (MdA) and site 2 (AdB) in 2021 and 2022. Bars with different lowercase letters differ significantly across treatments ($p < 0.05$).

The volume of standing biomass in site 2 was somewhat expected in 2022. However, site 1 standing biomass decreased by 26% from 2021 to 2022. These results for site 1 were unexpected but may have been due to annual precipitation. Rainfall in Year 2 was significantly lower than in Year 1, which may have reduced seed germination and plant growth during the optimum growing season. Differences in the date of biomass collection between Years 1 and 2 could potentially have impacted results as well. Biomass was collected in Year 1 approximately one month later than in Year 2. The date change was implemented in Year 2 to increase the efficiency of the organic herbicides by applying treatments at an earlier growth stage. Although this helped with herbicide efficiency, younger plants would result in less available biomass for site 1 (33 kg ha$^{-1}$ in 2021 and 24 kg ha$^{-1}$ in 2022), and site 2 yielded the greatest biomass (41 kg ha$^{-1}$) compared with site 1, which may be a result of increased weed species present at the time of data collection. Site MdA included 17 weed species, while site AdB had 19 weed species. Biomass data were not collected in Year 1 for this site, so there is no available information to determine if biomass differed between years.

The results of this study found that plot suppression was not impacted significantly by organic or synthetic herbicides across both years ($p = 0.314$, Figure 5). This would suggest that despite the different approaches in herbicide applications, Year 1 and Year 2 were comparable in overall herbicide efficiency per plot, suggesting weed populations were not significantly affected by initial herbicide applications. The time before the next spray application (for site MdA, 13 days later, and site AdB, 8 days later) would allow weed populations to recover and mature past the targeted growth stages recommended by herbicide labels; organic applications should be made in 10-to-14-day intervals as necessary to achieve acceptable control [22]. For these mature plants, weed physiology, weed morphology, vegetative mass, and spray retention are factors that would reduce herbicide performance throughout the experiment [23]. Weed physiology (absorption, translocation, and metabolism) would be lower due to leaf surfaces being heavily covered in wax and hairs. Weed morphology would not facilitate chemical penetration due to changes in leaf cutin, wax, cellulose, metabolites in the cell sap, and plant tissue. Vegetative

mass would prevent total coverage of an herbicide application and would require a higher volume of spray solution to kill target weeds. Spray retention is lower due to late-growth plants having a completed leaf structure compared to younger plants.

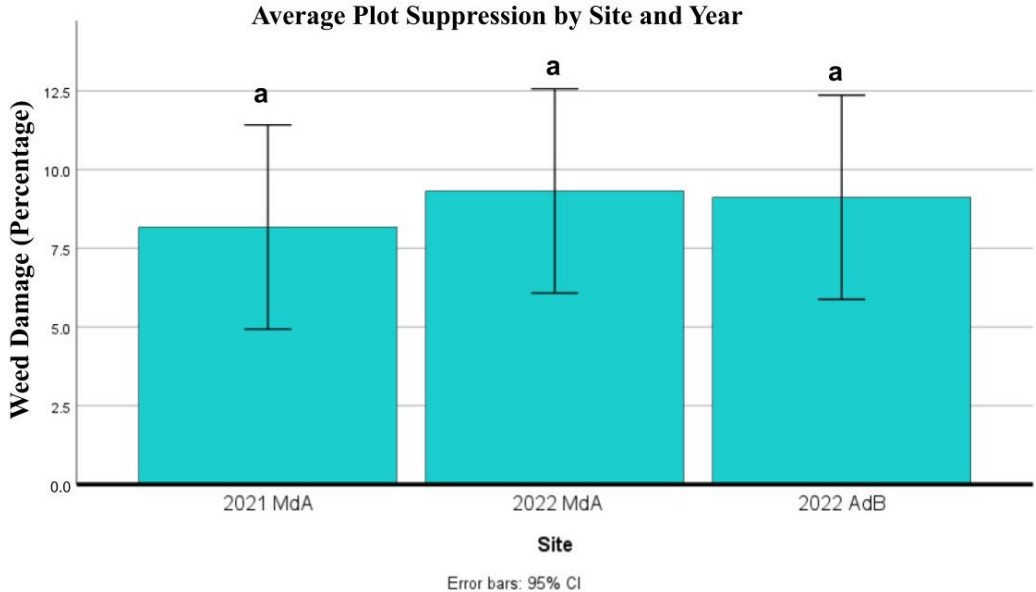

**Figure 5.** Average damage percentage rating (0–100%) at site 1 (MdA) and site 2 (AdB) in 2021 and 2022. Bars with different lowercase letters differ significantly across treatments ($p < 0.05$).

Roundup and Rifle herbicides were found to have the greatest impact on overall weed suppression across Year 1 ($p < 0.001$, Figure 6) and Year 2 ($p < 0.001$, Figure 7). As systemic herbicides work well on perennial and annual weeds, this may explain why they had the most effect. Contact herbicides are most often used to control annual weeds. However, contact herbicides that are applied to perennial weeds only kill the shoot, leaving the roots to re-sprout [24]. As such, perennial weed species can be controlled by contact herbicides but only after repeated chemical applications.

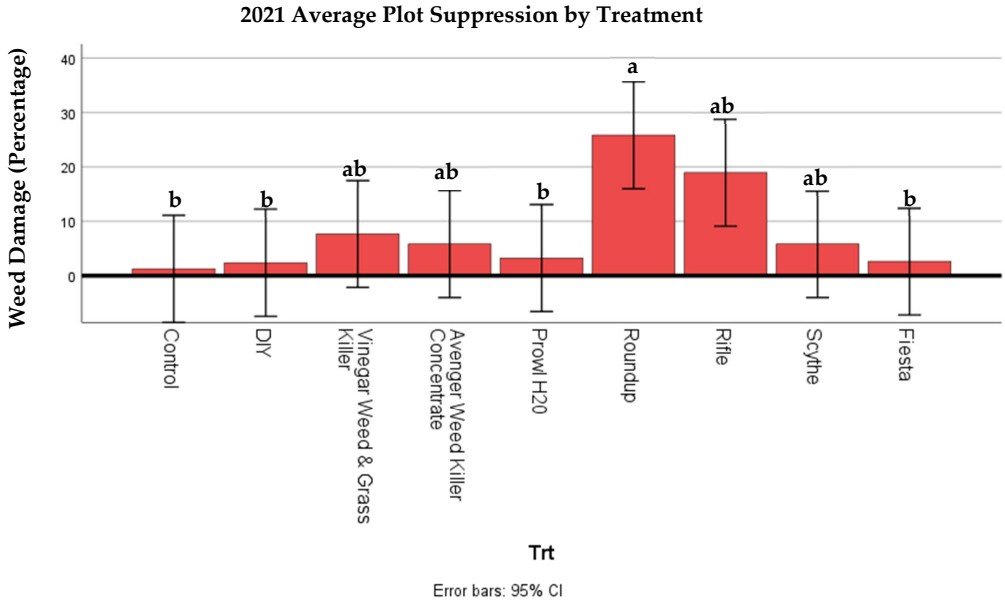

**Figure 6.** Average damage percentage rating (0–100%) for organic and synthetic herbicides in 2021. Bars with different lowercase letters differ significantly across treatments ($p < 0.05$).

**2022 Average Plot Suppression by Treatment**

**Figure 7.** Average damage percentage rating (0–100%) for organic and synthetic herbicides in 2022. Bars with different lowercase letters differ significantly ($p < 0.05$).

When reviewing the synthetic herbicides used, glyphosate, the active ingredient in Roundup, is a non-selective, systemic herbicide that moves from the treated foliage to other parts of the target plant. It acts as an inhibitor of the 5-enolpyruvylshikimate-3-phosphate synthase, stopping the mechanisms of the shikimic acid pathway and preventing the target plant from growing [25]. Rifle, a systemic herbicide, functions as a plant growth regulator. It accumulates in areas with phytohormones (auxins) and affects cellular elongation, turgor, cellular differentiation, and division [26]. The remaining synthetic herbicides did not have a significant impact on weed species. Prowl $H_2O$, a selective herbicide, utilizes a water-based formulation to maximize residual activity to prevent seeds from germinating. It acts as a meristematic inhibitor that interferes with plant cellular division or mitosis. However, it does not affect current live weed populations [27]. This could explain Prowl $H_2O$'s poor performance in Year 1 and Year 2, preventing new populations from emerging but not controlling existing populations. Fiesta, a selective herbicide, causes iron toxicity (oxidative damage) at a cellular level. Broadleaf plants are specifically targeted because of the iron absorption pathways in these types of plants, leaving all other types of plants unharmed [28]. This could explain Fiesta's lack of long-term control over the study. Rush, grass, and sedge weed species were left unaffected, allowing them to mature, seed, and populate the rest of the experimental plots. There is also the possibility that broadleaf weed species were not controlled after initial spray applications and grew beyond the optimum size for treatment [29].

Organic herbicides outperformed Fiesta and Prowl $H_2O$ herbicides in overall plot suppression but were not comparable to Roundup and Rifle applications. The DIY and Vinegar Weed & Grass Killer herbicides were the only organic treatments to have significantly improved in plot suppression percentages across years. This lack of improved weed control for all organic herbicides was unexpected, given that application rates increased in Year 2 to achieve maximum weed suppression potential (multiple applications, correct temperature, and timing of the day, higher volume of chemicals applied, etc.). This could be due to an organic herbicide's mode of action, additional environmental conditions not met, biomass amount, and lack of other organic management practices to assist in weed management. The DIY treatment, a contact, non-selective herbicide mixture, consists of

table salt, vinegar, and dish soap (triclosan). This combination of household ingredients strips the natural oils from the cuticular layer on plant leaves with dish soap and then further desiccates and dissolves the plant cell membranes with salt and vinegar [30]. However, this homemade herbicide has not been federally approved or tested for residential or commercial use. One possibility for this treatment not effectively controlling target weed species is the type of vinegar used. The average household vinegar contains 5% acetic acid, while organic vinegar herbicides utilize 20% acetic acid mixtures. In this case, the concentration of acetic acid used is not high enough to effectively kill a plant. Vinegar Weed & Grass Killer, a non-selective herbicide, utilizes a 20% acetic acid formulation to destroy and/or dissolve cell membranes, drying out plant tissue [31]. Similar to the DIY treatment, the Vinegar Weed & Grass Killer improved in performance across the years but did not sustain control of weed populations. This could be due to weed species recovering over time. The two remaining organic herbicides, Avenger Weed Killer Concentrate and Scythe, were comparable in performance across both years despite having different modes of action. Avenger Weed Killer Concentrate, a contact, non-selective herbicide, utilizes a citrus extract that removes the waxy cuticular layer from leaves, drying out plant tissue [32]. Scythe, a contact, non-selective herbicide, damages cell membranes by lowering the internal pH of plant cells. The result is cell leakage and death of all contact tissues [33].

The results found here are almost comparable to other studies concerning organic herbicides. Shresha et al. (2013) [34] stated mechanical cultivation outperformed and was more cost-effective than steam and d-limonene herbicide treatment applications when suppressing weed control. They evaluated four different organic treatments (French plow, bezzerides tree and vine cultivator, steam, and d-limonene herbicide) on organic raisin and wine grape vineyards in California. A French plow and the bezzerides tree and vine cultivator are both mechanically drawn farming attachments that disrupt the soil and potentially kill target weed species by exposing their root system and/or damaging the target plant [35]. High-temperature steam can kill weed species through the coagulation of proteins and burst cell walls through water expansion [36]. A d-limonene organic herbicide was used as well, similar to this study. The experimental design consisted of a split-plot design, with the aforementioned treatments and additional weed control treatments (hand hoeing, no hand hoeing in the raisin grape vineyards and hand hoeing, no hand hoeing, steam, and d-limonene herbicide in the wine grape vineyard) one month after the main treatments were applied. The results found that the use of organic herbicides (with multiple applications at specific sites) did not result in sufficient weed control. They state that the d-limonene herbicide and steam applications contributed to only 2–3 weeks of control in all three site locations, while the mechanical treatments provided up to 4 weeks of control. Comparatively, this study also found organic herbicide treatments to display limited long-term control over weed species populations. The results could be due to the level of disruption mechanical treatments were able to inflict throughout the study. Steam and d-limonene herbicides, both contact herbicides in nature, could have potentially affected only above-ground biomass, leaving root systems in place to recover at a later time.

In another study evaluating organic herbicides, Ferguson (2004) [37] reported inconsistent weed control with citric acid, clove oil, and thyme/clove oil herbicides when compared to glyphosate. They evaluated three site locations in southern and north central Florida from September to October 2003. Treatments were applied at the recommended concentration rates (undiluted citrus acid at 61 L ha$^{-1}$, clove oil at 76 L ha$^{-1}$, clove oil/thyme oil at 76 L ha$^{-1}$) and twice the recommended concentration rates to broadleaf and grass species. These weed species varied by location but consisted of Alexander grass, Bahia grass, Bermuda grass, carpetweed, crabgrass, hairy indigo, lamb's quarters, Florida pusley, goatweed, nutsedge, pigweed, shrubby primrose willow, broadleaf signal grass, southern sandbur, spurge, torpedo grass, and citrus rootstock seedlings. The results found the treatments were significant. Organic herbicides, at recommended and at higher concentrations, display weed control rates of 10 to 40% compared with the 100% control with glyphosate. Similarly, this study found comparative results in the efficacy of the organic

treatments used despite multiple applications and higher application rates. This could have been due to the growth stage of the weed species at each of the three site locations, which is more than the recommended application rates. The rates applied were greater than what was applied in this study, but if the growth stages of the weed species were larger than six inches, these contact herbicides could have burned the surface area of the target plants. Leaving the root system intact to recover later. The pounds per square inch (PSI) of the herbicide applications could also have influenced the overall coverage of the organic herbicides, further affecting herbicide efficacy.

In another organic herbicide study, Lanini (2023) [16] reported that the growth stage of a target weed species could impact the efficiency of an organic herbicide treatment on overall weed suppression. The organic herbicides used included Weed Pharm (20% acetic acid), C-Cide (5% citric aid), GreenMatch (55% d-limonene), Matratec (50% clove oil), WeedZap (45% clove oil + 45% cinnamon oil), and GreenMatch EX (50% lemongrass oil). Weed Pharm is a non-selective, natural organic herbicide that utilizes a 20% acetic acid base to rupture plant cells, causing fluids to leak and plant tissue to dry out [38]. C-Cide is a natural vitamin-C-based organic herbicide that passes into the stoma of the plant, travels through the xylem, "melts" the plant's cell walls, and prevents the plant from transporting vital nutrients, killing the plant [39]. GreenMatch is a broad-spectrum weed killer that contains citrus extract and controls most annual and perennial weeds by removing the waxy cuticle of the target plant [40]. Matratec is a contact, non-selective organic herbicide that utilizes clove oil to control annual and perennial broadleaf and grassy weeds by removing the waxy cuticle of the plant leaf [41]. WeedZap is a non-selective organic herbicide that kills broadleaf and grass weeds by dissolving plant cell wall structures [42]. GreenMatch EX is a non-selective organic herbicide that contains lemongrass oil, controlling annual and perennial broadleaf and grassy weeds by removing the waxy cuticle of the target plant [43] Weed Pharm, GreenMatch, Matratec, and GreenMatch EX's mode of actions were similar to the treatments used in this study, Avenger Weed Killer Concentrate and Scythe herbicides. The descriptions found for the C-Cide and WeedZap herbicides as a burndown of the plant cell structure would not be similar to any of the treatments used in this study. Broadleaf and grass weed species were evaluated in a greenhouse and field setting, with broadleaf weed species including pigweed and black nightshade and grasses including barnyardgrass and crabgrass. Herbicide applications were applied at 12 days, 19 days, and 26 days after weed emergence. The results found were significant when evaluating the growth stage of target weed species and organic herbicide applications. They saw that when applied at high volumes, organic treatments displayed 60% to 100% weed control on broadleaf weeds that were 12 days old. However, for broadleaf weeds, 26 days or older, weed control dropped to less than 40%. Grass species, even those at 12 days old, only displayed around 40% weed control throughout the study. Comparatively, this study found similar results when evaluating the effects of the treatment applications over the time evaluations, with the 24 h and 7 DAT evaluations having the best plot suppression percentages. This could have been due to the vulnerability of younger plants to herbicide applications versus mature plants. As mature plants have a well-established waxy cuticle and growth system, they are more likely to resist contact herbicide effects.

Despite the different modes of action for most of the organic herbicides used in this study, all of the organic treatments are contact only. In this way, the overall control of the target weed species was dependent on the type and amount of weed species present. Annual weeds could die after spray applications, but the roots of perennial weeds would remain alive, allowing them to repopulate. Sedge weed species are not as affected by contact herbicides, which allows these species to recover. Vegetative mass and environmental conditions during spray applications could have influenced lasting results, both preventing effective coverage and/or herbicide efficiency from occurring. Weed management is also something to consider with these results. Organic herbicides are not usually applied as the sole source of control in an organic farming operation. Other resources and methods applied, such as mechanical tillage, cover crops, crop rotation, mulches, and tarps, are

utilized together within a management program to control weed populations. Having little long-term control may be expected for these types of herbicides when evaluating the overall system put in place at these farming operations.

### 3.2. Weed Species

Organic and synthetic herbicide performance and weed suppression differed by treatment and by timing, with 7 DAT < 24 h < 30 DAT < 60 DAT ($p < 0.001$. Table 7). However, weed species suppression ratings were comparable across years and site locations ($p = 0.785$). This would suggest that while weed species suppression was highest in the 7 DAT to 30 DAT timed evaluations, the overall weed suppression ratings were similar across years and sites. These results are surprising. However, organic herbicides showed significant damage/and or weed suppression in the first few days following a spray application, but the target weed populations would recover in about two weeks [44]. In Year 1, seven days after the initial herbicide application, it is likely weed species recovered or repopulated experimental plots before the assessment of the next timed evaluation (30 DAT). In Year 2, multiple applications occurred prior to the 30 DAT and 60 DAT, but environmental conditions (microbial degradation activity or time lapse between sprayings) could have allowed weed populations to recover or not sustain permanent damage leading to complete death. These results could suggest that initial activity on target organisms displays some specific weed suppression but that, over time, weed suppression decreases in all treatments.

**Table 7.** Average herbicide efficacy on weed species measured as percent damage at 24 h, 7 days, 30 days, and 60 days after application. Data were collected at site 1 (MdA) in 2021 and 2022 and site 2 (AbB) in 2022. Values were measured by a 0–100 rating scale, with 0 indicating no damage and 100 indicating complete weed destruction.

| Treatments | Year 1 | | | | Year 2 | | | | | | | |
| | MdA Damage % | | | | MdA Damage % | | | | AdB Damage % | | | |
| | 24 H | 7 DAT | 30 DAT | 60 DAT | 24 H | 7 DAT | 30 DAT | 60 DAT | 24 H | 7 DAT | 30 DAT | 60 DAT |
|---|---|---|---|---|---|---|---|---|---|---|---|---|
| Control | 0% | 0% | 0% | 0% | 0% | 0% | 0% | 0% | 0% | 0% | 0% | 0% |
| DIY | 5% | 4% | 0% | 0% | 7% | 7% | 0% | 0% | 11% | 11% | 0% | 0% |
| Vinegar Weed & Grass Killer | 15% | 15% | 0% | 0% | 10% | 6% | 0% | 0% | 16% | 34% | 0% | 0% |
| Avenger Weed Killer Concentrate | 11% | 12% | 0% | 0% | 8% | 7% | 0% | 0% | 8% | 14% | 0% | 0% |
| Prowl H$_2$O | 5% | 8% | 0% | 0% | 5% | 8% | 0% | 0% | 8% | 10% | 0% | 0% |
| Roundup | 7% | 96% | 0% | 0% | 9% | 81% | 0% | 86% | 15% | 81% | 0% | 0% |
| Rifle | 10% | 66% | 0% | 0% | 12% | 66% | 0% | 0% | 8% | 63% | 0% | 0% |
| Scythe | 10% | 13% | 0% | 0% | 9% | 8% | 0% | 0% | 11% | 20% | 0% | 0% |
| Fiesta | 4% | 7% | 0% | 0% | 4% | 5% | 0% | 0% | 8% | 13% | 0% | 0% |

DAT—days after treatment, DIY—Do it yourself.

For 24 h after initial applications and 7 DAT, some significant differences were found between specific treatments and weed species, both in Year 1 (Table 8) and Year 2 (Tables 9 and 10). For Year 1, dog fennel was found during the 24 h and 7 DAT evaluations, but only within a few plots. Dallisgrass, chickweed, hedge hyssop, and valley redstem were found in plots during the 30 DAT evaluation. Alligator weed and grasslike fimbry were found in plots during the 30 DAT and 60 DAT evaluations. As such, a significant weed species suppression rating by treatment could not be estimated for these weed species as there were limited data to evaluate. Concerning the weed species that were present during the initial spray application, Roundup was the most effective in controlling weed species. However, for nutsedge and doveweed, Rifle and Vinegar Weed & Grass Killer herbicides, respectively, were comparable to Roundup in suppression. These results could be explained by weed type and growth stage at the time of application. The weed nutsedge

is a sedge that is typically more controlled by systemic herbicides like Roundup and Rifle than other contact herbicides. Doveweed is a broadleaf weed. However, if it was in an early growth stage or was able to receive full coverage during a spray application, the organic herbicide Vinegar Weed & Grass Killer could have been effective in suppressing this species.

**Table 8.** Year 1 herbicide performance by weed species. Data were collected at site 1 (MdA) in 2021. Values were measured at a 0–10 rating scale, with 0 indicating no damage and 10 indicating complete weed destruction.

| Treatment | NS | LB | BWH | DW | BG | CYW | PC | SH | NA | JM |
|---|---|---|---|---|---|---|---|---|---|---|
| Control | 1.38 c | 0.50 c | 1.50 c | 0.38 c | 1.50 cb | 0.00 b | 0.00 b | 1.00 b | 1.60 c | 0.00 b |
| DIY | 3.00 bc | 1.75 bc | 1.88 bc | 1.25 c | 0.25 c | 0.75 b | 0.00 b | 1.75 ab | 0.00 e | 0.29 b |
| Vinegar Weed & Grass Killer | 3.40 b | 3.60 ab | 1.40 c | 6.20 a | 3.75 abc | 0.00 b | 1.75 ab | 2.50 ab | 0.00 e | 0.75 b |
| Avenger Weed Killer Concentrate | 3.25 b | 2.38 bc | 2.38 bc | 2.38 bc | 4.25 ab | 0.50 b | 1.75 ab | 1.50 ab | 0.00 e | 2.00 ab |
| Prowl H$_2$O | 3.13 b | 2.63 bc | 2.88 bc | 1.75 c | 2.20 bc | 1.20 ab | 1.40 ab | 3.00 ab | 0.00 e | 0.00 b |
| Roundup | 5.62 a | 5.88 a | 6.50 a | 6.13 a | 7.00 a | 5.00 a | 6.00 a | 6.00 a | 6.33 a | 5.00 a |
| Rifle | 5.33 a | 4.00 ab | 4.67 ab | 4.83 ab | 3.67 abc | 2.00 ab | 0.00 b | 4.00 ab | 4.00 b | 0.71 b |
| Scythe | 3.50 b | 1.63 bc | 2.38 bc | 2.38 bc | 2.67 bc | 2.33 ab | 0.33 b | 0.67 b | 1.17 c | 0.88 b |
| Fiesta | 2.00 bc | 1.37 cb | 2.75 bc | 1.13 c | 2.00 bc | 2.00 ab | 1.75 ab | 1.00 b | 0.50 d | 0.00 b |

NS—nutsedge; LB—lawn burweed; BWH—blue water hyssop; DW—doveweed; BG—barnyard grass; CYW—common yellow woodsorrel; PC—purple cudweed; SH—spreading hedge; NA—narrowleaf aster; JM—Japanese mazus. A compact letter display is used to show comparisons between the different treatments within the table. Where *a* is significantly different from *b*, but *ab* is not significantly different from *b*.

**Table 9.** Year 2, site 1 (MdA) herbicide efficiency by weed species. Data were collected at site 1 (MdA) in 2022. Values were measured at a 0–10 rating scale, with 0 indicating no damage and 10 indicating complete weed destruction.

| Treatment | NS | JR | CYW | PC | SH | IA | TG | AW | WB |
|---|---|---|---|---|---|---|---|---|---|
| Control | 0.00 b | 0.00 a | 0.00 b | 0.29 a | 0.00 c | 0.00 a | 0.00 b | 0.07 b | 0.00 b |
| DIY | 0.94 b | 0.33 a | 1.42 ab | 1.60 a | 2.00 bc | 1.60 a | 0.78 b | 0.29 ab | 2.00 b |
| Vinegar Weed & Grass Killer | 1.25 b | 0.00 a | 1.27 ab | 1.57 a | 1.55 bc | 1.25 a | 0.00 b | 0.42 ab | 1.50 b |
| Avenger Weed Killer Concentrate | 0.88 b | 0.10 a | 2.09 ab | 1.16 a | 1.55 bc | 0.92 a | | 0.57 ab | 2.13 ab |
| Prowl H$_2$O | 1.00 b | 0.45 a | 1.27 ab | 1.33 a | 1.43 bc | 1.36 a | 0.00 b | 0.33 ab | 1.25 b |
| Roundup | 4.69 a | 0.38 a | 4.00 a | 4.71 a | 5.22 a | 2.40 a | 3.00 a | 1.75 a | 5.13 a |
| Rifle | 1.63 b | 0.00 a | 3.18 a | 4.33 a | 3.40 ab | 2.18 a | 0.00 b | 1.15 ab | 2.63 ab |
| Scythe | 0.81 b | 0.20 a | 1.67 ab | 3.00 a | 2.22 bc | 1.20 a | 0.33 b | 0.00 b | 1.78 b |
| Fiesta | 0.69 b | 0.33 a | 1.18 ab | 1.13 a | 1.50 bc | 1.40 a | 0.00 b | 0.43 ab | 1.33 b |

NS—nutsedge; JR—jungle rice; CYW—common yellow woodsorrel; PC—purple cudweed; SH—spreading hedge; IA—Iva annua; TG—tall goldenrod; AW—alligator weed; WB—western buttercup. A compact letter display is used to show comparisons between the different treatments within the table. Where *a* is significantly different from *b*, but *ab* is not significantly different from *b*.

For weed species that germinate after initial herbicide applications, postemergence herbicides must be sprayed in accurate doses so that the amount taken into the crop plant will not be enough to cause injury [45]. Late herbicide applications must take into account how mature the crop plant/weed species is, what herbicide is being used, and/or how many days there are before harvest. Along with this, most herbicides have cutoff restrictions, and growers must consider potential crop damage and yield loss from sprayer wheel tracks [46]. Other considerations include weed species' effect on crop yield. Weed species that appear after crop plants are well established are potentially less likely to compete for nutrients, light, water, and space so weed control may not be needed in this instance. As such, growers typically do not apply late-season herbicide applications as they are unlikely to provide significant benefits for most fields [47].

**Table 10.** Year 2, site 2 (AdB) herbicide efficiency by weed species. Data were collected at site 2 (MdA) in 2022. Values were measured at a 0–10 rating scale, with 0 indicating no damage and 10 indicating complete weed destruction.

| Treatment | BWH | TG | BC | WC | GR | PEP | PWG | LW | SP |
|---|---|---|---|---|---|---|---|---|---|
| Control | 0.33 a | 0.60 a | 0.00 a | 0.00 a | 0.13 b | 0.00 a | 0.00 a | 1.17 a | 0.43 a |
| DIY | 3.00 a | | 2.50 a | 1.60 a | 1.86 ab | 0.67 a | 0.56 a | 1.09 a | 1.00 a |
| Vinegar Weed & Grass Killer | 2.60 a | 0.33 a | 2.50 a | 1.00 a | 3.67 ab | 2.14 a | 0.58 a | 0.93 a | 1.25 a |
| Avenger Weed Killer Concentrate | 2.00 a | 0.50 a | 2.40 a | 0.22 a | 3.00 ab | 2.00 a | 1.17 a | 1.17 a | 1.20 a |
| Prowl H$_2$O | 1.25 a | 0.44 a | 2.67 a | 1.00 a | 1.60 ab | 1.00 a | 0.38 a | 1.00 a | 0.40 a |
| Roundup | 5.17 a | 1.50 a | 4.50 a | 1.88 a | 5.75 a | 1.43 a | 0.69 a | 1.92 a | 3.71 a |
| Rifle | 4.25 a | 0.00 a | 2.50 a | 1.30 a | 4.71 a | 1.30 a | 0.75 a | 2.14 a | 3.14 a |
| Scythe | 2.50 a | 0.00 a | 4.33 a | 1.30 a | 3.25 ab | 1.57 a | 0.64 a | 2.47 a | 0.63 a |
| Fiesta | 2.75 a | 0.60 a | 1.00 a | 0.75 a | 2.71 ab | 1.75 a | 0.29 a | 0.93 a | |

BWH—blue water hyssop; TG—tall goldenrod; BC—burr clover; WC—white clover; GR—grassleaf rush; PEP—pink evening primrose; PWG—Paraguayan windmill grass; LW—longleaf wedgescale; SP—scarlet pimpernel. A compact letter display is used to show comparisons between the different treatments within the table. Where *a* is significantly different from *b*, but *ab* is not significantly different from *b*.

For Year 2, site 1, lawn burweed and doveweed were found during the 24 h, 7 DAT, and 30 DAT evaluations. However, treatment applications reduced the number of lawn burweed throughout the study, and none were found at the 60 DAT. Blue water hyssop was found during each of the timed evaluations but reduced in number throughout the study. Dog fennel was found sporadically between the 7 DAT and 30 DAT evaluations. Dallisgrass, grasslike fimbry, chickweed, and hedge hyssop were found during the 30 DAT and 60 DAT evaluations. Dandelion and spiny sowthistle were only found during the 30 DAT evaluation. Narrowleaf aster was found throughout the study and was not controlled by the herbicide treatments. Fox tail was found in the 60 DAT evaluation. As such, a significant damage rating by treatment could not be estimated for these weed species, as either there were limited data to evaluate or ratings were 0 and could not be statistically run. Concerning the weed species that could be evaluated, Roundup was the most significantly effective in controlling weed species. However, for this site, the common yellow woodsorrel was equally affected by Roundup and Rifle. As these are both systemic herbicides, this could explain any significance.

For Year 2, site 2, lawn burweed and spiny sowthistle were found during each of the timed evaluations but were reduced in number throughout the study. Doveweed, Iva annua, chickweed, and western buttercup were only found during the 24 h evaluation. Jungle rice, dandelion, and spotted spurge were found during the 30 DAT and 60 DAT evaluations and were not affected by treatment applications. Horseweed was found only during the 60 DAT evaluation. Common yellow woodsorrel was found throughout the study, but there was no long-term control for this species. Purple cudweed was found throughout the study, but the population was reduced by 60 DAT. American burnweed and southern crabgrass were found in each evaluation but were sporadic in number. As such, a significant damage rating by treatment could not be estimated for these weed species, as either there were limited data to evaluate or ratings were 0 and could not be statistically run. Concerning the weed species that could be evaluated, Roundup was the most significantly effective in controlling weed species. However, for this site, the grassleaf rush was equally affected by Roundup and Rifle. As these are both systemic herbicides, this could explain any significance.

## 4. Discussion and Future Research

Weed populations can be detrimental to crop plants through competition, reducing crop yield, and hosting pests and diseases. The use of herbicides is one of the most efficient and cost-effective methods for controlling weeds in weed management programs. However, the use of organic herbicides in conventional cropping systems is not realistic at the present

time. Despite different approaches to control and different modes of action, the organic herbicides in this study were statistically similar in performance. Synthetic herbicides, specifically Roundup and Rife herbicides, displayed the highest level of weed control across years and site locations. Other contact herbicides were ineffective on many of the weed species, displaying some physical damage, but most species recovered over time. Weed species sensitivity to both organic and synthetic herbicides showed a linear relationship with specific species but were not significant past the 7 DAT evaluation. This suggests that after the first week, weed populations and biomass recovered and were not different from control plots. Without the presence and competition of row crops usually associated with farming operations, fallow fields quickly reformed back to existing conditions prior to spray applications. According to the results of this study, under specific environmental conditions, organic herbicides did not present adequate control and residual to sustain weed suppression in fallow fields prior to row crop planting.

Moving forward, changes in experimental design, regional locations, and advances in precision spraying technology could help address factors that were masked by field variables. Soil moisture and precipitation, specifically, were different in 2021 and 2022, potentially limiting herbicide persistence and affecting the volume of biomass found between years for site 1. Research to address field variability may include replication of this study in other soil types and climate conditions, in addition to expanding the research from fallow to row crop fields. Increasing knowledge of organic herbicide treatments on weed suppression in different environments could be beneficial not only to the producer seeking alternative control methods but potentially minimize soil health impacts. Replicating this study in a greenhouse setting or outside container experiment could improve treatment foliage cover on weed species, improve weed species control with specific herbicides, control soil moisture levels, and eliminate potential biomass overlap. Some concerns with working in a greenhouse or container experiment setting could be the potential for pests or disease outbreaks and maintenance/monitoring expenses. Advances in spray technology, specifically in precision spraying software like WEED-IT (Version 2.0) [48], can be utilized to spot spray living plant populations in a fallow field. Reducing the need for blanket applications and potentially increasing the efficacy and viability of using organic herbicides in a conventional setting.

Future research to include new herbicide technologies and herbicide mix combinations is needed in order to produce efficient organic herbicide recommendations for conventional use. Improvements in organic herbicide technologies could include developing modes of action through the utilization of new novel natural substances (herb oil extracts, natural acids, etc.) for cell membrane disruption or creating systemic organic herbicide options. Organic herbicides' only mode of action is through contact, and systemic pathways could potentially increase their efficiency, making them more viable for use in a commercial setting. Other technology options could include the development of natural herbicides with soil residual activity. This could be achieved by creating soil-applied organic herbicides with complex chemical structures, inherently slowing microbial degradation rates of these chemicals. This could potentially increase herbicide persistence and help control weeds after initial applications. New herbicide mixtures could increase the efficacy of organic herbicides while sustaining what is available in the current market.

**Author Contributions:** Conceptualization, L.F. and C.d.; methodology, L.F., and C.d.; validation, A.H., C.d. and L.F.; formal analysis, C.d.; resources, L.F.; writing—original draft preparation, C.d.; writing—review and editing, A.H. and L.F.; project administration, L.F. and A.H.; funding acquisition, L.F. All authors have read and agreed to the published version of the manuscript.

**Funding:** This research was funded through Hatch funding project no. LAB74293.

**Institutional Review Board Statement:** Not applicable.

**Informed Consent Statement:** Not applicable.

**Data Availability Statement:** Data are contained within the article.

**Conflicts of Interest:** The authors declare no conflicts of interest.

**Disclaimer:** Mention of trade names or commercial products in this publication is solely for the purpose of providing specific information and does not imply recommendation or endorsement of by Louisiana State University Agricultural Center.

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
