# Peer review of "Evaluation of Organic and Synthetic Herbicide Applications on Weed Suppression in a Conventional Cropping System in Louisiana"

_sustainability, doi:10.3390/su16073019_

Round 1

Reviewer 1 Report

Comments and Suggestions for Authors

In this study, the efficacy of organic and synthetic herbicides in weed suppression was evaluated through a comprehensive survey. The experimental design encompassed nine treatments, each replicated four times and conducted on two distinct soil types. Results indicated a notable difference between the weed suppression capabilities of organic and synthetic herbicides. The article emphasizes the importance of adhering to recommended herbicide application rates and adopting environmentally responsible practices in weed management programs to mitigate potential environmental effects. Despite providing valuable insights into sustainable practices, the article has several areas that need attention:

1. The abstract identifies potential consequences of using synthetic herbicides but lacks clarity on key study challenges and motives.

2. Section 1's limited references necessitate an expanded literature review, particularly in areas related to sustainable agriculture and weed suppression.

3. A detailed flowchart in the experimental design and field management sections would enhance clarity.

4. Section 2.5 requires clarification on statistical analysis procedures, and the experiment analysis should include a comparison of different statistical models.

5. Figures 3-6 need improvement in image quality and larger caption sizes for enhanced clarity.

6. Sections 4-5 could be consolidated into a single conclusion section.

7. The reference list should prioritize recent journal papers over websites, ensuring a more conventional citation format.

Reviewer 2 Report

Comments and Suggestions for Authors

The paper presents research on the evaluation of weed control in comparison with natural and synthetic herbicides. The experiment was well designed. The results are well described and discussed. The article may be interesting for readers. The reviewer made some minor comments for the authors:

·        Throughout the text - use superscripts. Replace „ha-1” with ha-1

·        Experimental Design and Field Management - On what basis was the herbicide dose changed exactly in the second year of the experiment and what were the doses?

·        Figure 2 - apply error bars to the chart

·        Line 298 -  repalce „In fact, they state the d-limonene herbicide” with They state the d-limonene herbicide

·        Line 303 - Replace „over the course of the study” with throughout the study

·        Lines 315-316 „found that there was a significance between treatments” with found that treatments were significant

·        After the methods section, please justification the text in your work. Currently the text is left-aligned

·        Please add numerical values in the results section

Comments on the Quality of English Language

 Minor editing of English language required

Reviewer 3 Report

Comments and Suggestions for Authors

The manuscript summarizes results of the comparative study on the efficiency of synthetic and herbicides for weed suppression which were performed in the two-year experiments including nine treatments on two different soil types. The main conclusion on the inefficiency of organic herbicides comparing to the synthetic ones is based on a set of valuable experimental data and their adequate treatment using adequate statistical methods.

 The topic of the research is relevant. Weed control problem is a central one in modern crop production. Herbicide treatments are ubiquitously applied to solve the problem. However, the knowledge on the effects of different classes of the substances, the synthetic and organic ones, on different weed species is still insufficient. The limited number of studies is available, and they consider only limited number of substances and limited number of plant species. This prevents elaborating recommendations for herbicide utilization in rational plant growing. The study performed provides valuable and statistically confirmed data.

 The title of the manuscript corresponds to the content.

 The design of the experiment is appropriate to the main goal of the research, namely to the evaluation of efficiency of the two classes of herbicides.

 The section “Future research” highlights the proposed ways of management of different classes of herbicides intended to weed control.

 The manuscript is clearly written and sufficiently illustrated.

 Meanwhile I have some comments and recommendations.

 1.     General comment:

 Weed classification terminology should be carefully revised throughout the text. The generally accepted weed classifications use different criteria. According to morphological characters, weeds are classified as grasses, sedges, and broadleaf weeds. Classification according to the life cycle distinguishes annual, biennial, and perennial weeds. Therefore, this should be clearly designated when you use, for example, the terms based on morphological characters and life cycle. For example: “Products containing vinegar (i.e. acetic acid) are more effective in controlling grasses and annual weed species than broadleaf and perennial weeds” (lines 57-59). Or: “Annual weeds could die after spray applications but the roots of perennials and grasses…” (line 367) – grasses also can have perennial life cycle.

 I recommend to include information on the generally accepted classification of weeds in the Introduction section. This will help to better understand interpretation of the results. In this relation, I also recommend to revise the content of tables 3 and 4: 1) to change the name of the first column from “Weed species” (that is actually the in the third column) to “Common name”; 2) to remove the second column from the table 3 (the content is clear from the table name); 3) ) to change the forth column “Leaf type” to “Growth forms”; 4) to change the name of the last column “Growing season” to “Life cycle”.

 2.     Notes to the section “Materials and Methods”:

- As follows from the “Materials and Methods” section, the experimental fields “were located approximately 14.5 km apart” (line 84). Figure 2 represents the average minimum temperatures and precipitations for both. Does it mean that the weather conditions were completely identical at both sites? It is necessary to specify.

- A question arose on the methodology applied for estimation of weed suppression. Is the scale for estimating weed suppression original? When describing estimation of weed survival, the authors cite the article (Waryszak et al. Herbicide effectiveness in controlling invasive plants under elevated CO2: sufficient evidence to rethink weeds management. J. of Environ. Manag. 2018, 226, 400-407). Citation from the manuscript (Lines 152-155): “Herbicide efficacy for weed control and suppression was evaluated for each weed species. This was achieved using a 0-10 rating scale, with zero indicating no damage and ten indicating complete weed destruction (Table 6). This is a cost-efficient, common method for rating herbicide toxicity (Waryszak et al., 2018). However, the scale of rating proposed in the table 6 of the manuscript is absent in the citing article. There is the following statement in the article of Waryszak et al., 2018: “A plant was considered alive if it still had live tuber (A. cordifolia only) or leaf material (all other species). In this relation, I recommend to specify the scale used with indicating, for example, what does it mean such terms from table 6 as “symptoms”, “damage”, “suppression” etc. It is unclear the need for the presence of table 6 in the text since it does not contain information of the percentage plant damage.

Formal notes:

Table 5 First column of the table 5 head should be corrected.

The footnotes to the tables 8, 9, 10 contain the phrase “Bars with different lower-case letters” which is unnecessary in the content.

Round 2

Reviewer 3 Report

Comments and Suggestions for Authors

Authors have thoroughly revised the manuscript and presented a point by point response. I am satisfied.

The manuscript can be accepted in the present form.